**Data Availability Statement:** All relevant data are in the paper and its Supporting Information files.

**Funding:** This work was supported by grants from the Guangxi Natural Science Foundation Program

# Diagnostic accuracy of pelvic magnetic resonance imaging for the assessment of bone marrow involvement in diffuse large B-cell lymphoma

Qing Ke[1]☯, Cheng-Cheng Liao[1]☯, Xiao-Hong Tan[1], Bao-Ping Guo[1], Hong Cen[1]*, Le-Qun Li[2]*

1 Department of Hematology/Oncology, Guangxi Medical University Cancer Hospital, Nanning, China,
2 Department of Hepatobiliary Surgery, Guangxi Medical University Cancer Hospital, Nanning, China

☯ These authors contributed equally to this work.
* cenhong0771@163.com (HC); lequnli1110@163.com (LQL)

## Abstract

### Purpose

We investigated the efficacy of pelvic magnetic resonance imaging (MRI) in the diagnosis of bone marrow involvement (BMinv) in diffuse large B-cell lymphoma (DLBCL) patients.

### Patients and methods

This was a retrospective study of data from a previous study (NCT02733887). We included 171 patients who underwent bone marrow biopsy (BMB) and bone marrow smear (BMS), pelvic MRI, and whole-body positron emission tomography-computed tomography (PET/CT) from January 2016 to December 2019 at a single center. BMB/BMS and whole-body PET/CT results were used as reference standards against which we calculated the diagnostic value of pelvic MRI for BMinv in DLBCL patients. A chi-square test was used to compare detection rates, and a receiver operating characteristic curve was used to evaluate diagnostic value of pelvic MRI. Propensity-score matching was performed according to clinical information, and Kaplan-Meier curves were constructed to compare progression-free survival (PFS) and overall survival (OS) of patients.

### Results

The BMinv detection rate of pelvic MRI (42/171) was higher ($P = 0.029$) than that of BMB/BMS (25/171), and similar to that of PET/CT (44/171; $P = 0.901$). The sensitivity, specificity, accuracy, positive predictive value, and negative predictive value of pelvic MRI were 83.33%, 98.37%, 94.15%, 95.24%, and 93.80%, respectively. Median PFS values were as follows: BMB/BMS-positive, 17.8 months vs. BMB/BMS-negative, 26.9 months ($P = 0.092$); PET/CT-positive, 24.8 months vs. PET/CT-negative, 33.0 months ($P = 0.086$); pelvic MRI-positive, 24.9 months vs. pelvic MRI-negative, 33.1 months ($P < 0.001$). Median OS values were as follows: BMB/BMS-positive, 22.3 months vs. BMB/BMS-negative, 29.8 months

(2016GXNSFDA380029, 2018GXNSFBA281026, and 14124003-4), Fellowship of China Postdoctoral Science Foundation (2020M673555XB),the Self-financed Scientific Research Project of Guangxi Zhuang Autonomous Region Health and Family Planning Commission (Z2016504) and the Guangxi Science and Technology Foundation (GK2013-13-B-01). The funders had no role in study design, data collection and analysis, decision to publish, or preparation of the manuscript.

**Competing interests:** The authors have declared that no competing interests exist

($P$ = 0.240); PET/CT-positive, 27.9 months vs. PET/CT-negative, 33.9 months ($P$ = 0.365); pelvic MRI-positive, 27.3 months vs. pelvic MRI-negative, 35.8 months ($P$ = 0.062).

## Conclusion

Pelvic MRI is effective for detecting BMinv in DLBCL patients, providing a more accurate indication of PFS than BMB/BMS and PET/CT do. It may ultimately be used to improve the accuracy of clinical staging, guide patient treatment, and evaluate prognosis.

## Introduction

Diffuse large B-cell lymphoma (DLBCL) is the most common type of non-Hodgkin lymphoma [1], and approximately 11%–34% of these patients have bone marrow involvement (BMinv) at the time of diagnosis [2]. Bone marrow assessment forms part of the Ann Arbor staging system, and if a patient exhibits BMinv, the lymphoma is classified as stage IV [3–5], which has a high relapse rate and poor prognosis [1]. Bone marrow biopsy (BMB) is the main method used for clinical assessment of BMinv [6]. It is also used to evaluate marrow cellularity, hematopoietic reserve, and certain prognostic features. However, BMB is an invasive procedure with a relatively high false-negative rate due to inherent limitations in the biopsy site and the amount of tissue removed [7, 8].

In several studies in recent years, whole-body magnetic resonance imaging (MRI) and positron emission tomography-computed tomography (PET/CT) has been demonstrated to possess a higher diagnostic value than BMB in the detection of BMinv in lymphoma [9–11]. However, both PET/CT and whole-body MRI are relatively costly, which restricts their clinical application. Common sites of BMinv include the vertebrae, pelvis, and femur [12]. Although pelvic MRI is an effective method to screen the pelvis, the femur, and certain vertebrae, there have been no studies to assess its value in the diagnosis of BMinv in DLBCL. Therefore, a retrospective study was conducted to investigate the diagnostic value of pelvic MRI for BMinv in DLBCL, by comparing it with PET/CT and BMB. Moreover, the imaging characteristics of pelvic MRI and PET/CT were analyzed in terms of bone marrow lesion distribution. Furthermore, we evaluated their prognostic value through survival analysis. We demonstrated that pelvic MRI may not only be an alternative, but actually improve the BMinv detection rate for DLBCL patients.

## Material and methods

### Clinical data

Altogether, this retrospective study included 171 patients who received confirmation of a diagnosis of DLBCL at Guangxi Medical University Cancer Hospital in China from January 2016 to December 2019. The data was extracted from a previous study (NCT02733887) that we conducted. The inclusion criteria were as follows: (1) patients aged ≥18 years; (2) patients who were newly diagnosed; (3) patients diagnosed with DLBCL using pathological histomorphology and immunohistochemistry, according to the 2016 revision of the World Health Organization classification of lymphoid neoplasms [13]; (4) patients for whom BMB/bone marrow smear (BMS), pelvic MRI, and whole-body PET/CT were available; (5) patients followed up for ≥6 months; and (6) patients who received the R-CHOP-21 regimen. The latter consists of a once-off infusion of cyclophosphamide (750 mg/m$^2$), doxorubicin (50 mg/m$^2$), vincristine (1.4 mg/m$^2$, up to a maximum dose of 2 mg), and rituximab 375 mg/m$^2$ on day 1, followed by

oral prednisolone (40 mg/m$^2$ daily) on days 1–5, in 21-day cycles. All patients underwent BMB/BMS, pelvic MRI, and whole-body PET/CT examination within 1 week after confirmed pathological diagnosis.

This retrospective study was approved by the Ethics Commitment of Guangxi Medical University Cancer Hospital (LW2020028), and was conducted according to the principles expressed in the Declaration of Helsinki. Patients provided written informed consent before their data were analyzed.

## Examination equipment and methods

**Examiners.** In our study, for each examination method (BMB/BMS, PET/CT, and pelvic MRI), there were two different, experienced diagnosticians who determined independently whether there was BMinv. They were aware of the pathological diagnosis of the patient, but were blinded to each other's results and diagnoses. The physicians who used BMB/BMS for diagnosis were Dr. Ling-Sha Huang (18 years of diagnostic experience) and Dr. Mei-Qi Li (16 years of diagnostic experience). The physicians who used PET/CT for diagnosis were Dr. Guo-You Xiao (12 years of diagnostic experience) and Dr. Xin Zhao (11 years of diagnostic experience). The physicians who used pelvic MRI for diagnosis were Dr. Dong Xie (13 years of diagnostic experience) and Dr. Zheng Wang (12 years of diagnostic experience).

**Pelvic MRI.** Pelvic MRI was performed using the GE Discovery MR750w 3.0T system (GE Healthcare, Chicago, IL, USA), and the scans were completed with a body coil. Routine, T1-weighted image and fat-suppressed, T2-weighted image sequence scans were performed, each in both the coronal and transverse planes. The pelvic MRI scan field of view contains the ilium, the 3rd to 5th lumbar vertebrae, the sacrococcyx, ischium, pubis, and upper femur. Imaging parameters are indicated in Table 1. Lymphoma BMinv was defined using the following features of pelvic MRI [14]: the MRI signal intensity of bone marrow lesions is generally isointense or hypointense in comparison to the surrounding muscle and vertebrae on T1-weighted images, and hyperintense in comparison to the surrounding muscle on fat-suppressed T2-weighted images. The pattern of BMinv was classified as focal or diffuse.

**PET/CT.** Whole-body PET/CT was performed using a GE Discovery PET/CT 710 system (GE Healthcare) and $^{18}$F-fluorodeoxyglucose (FDG) synthesized with an HM-10 cyclotron (Sumitomo Heavy Industries, Ltd., Tokyo, Japan). Prior to scanning, patients were fasted for 6 h, and their blood glucose was verified to be less than 8.8 mmol/L. Subsequently, patients were injected with 4.44–5.55 MBq/kg body weight of FDG, rested for 50–60 min, and their bladder was emptied before scanning. Imaging included both a plain CT scan and a PET scan. All patients underwent non-contrast CT (tube voltage: 80 kV, current: 150 mA, slice thickness: 0.625 mm) followed by PET scans from the top of the skull to the middle of the femur, and, where necessary, to the sole of the foot. PET was performed for seven or eight bed positions, with 2.0 min per bed position. The workstation software was used to reconstruct the acquired data via the iterative attenuation correction method, to obtain transverse, sagittal, and coronal images, as well as a fusion image of the latter two. Focal bone marrow FDG uptake with or without increased diffuse uptake (higher than that in the liver) indicated positivity for BMinv [15].

**Table 1. Pelvic magnetic resonance image acquisition parameters.**

| Sequence | FOV (mm$^2$) | TR/TE/FA (ms/ms/deg) | Acquisition matrix (pixels) | Slice thickness/gap (mm) | ETL (echoes) | Scan time (min:s) |
|---|---|---|---|---|---|---|
| T1WI | 40 | 573/12/111 | 320×256 | 5/2 | 3 | 1:55 |
| T2WI | 38 | 5278/68/111 | 320×256 | 5/2 | 16 | 4:19 |

**Abbreviations:** ETL, echo train length; FA, flip angle; FOV, field of view; TE, echo time; TR, repetition time; WI, weighted image.

**BMB/BMS.** The anterior superior iliac spine/posterior superior iliac spine was routinely selected for BMB/BMS. BMB/BMS were performed by two pathological specialists at our hospital. Lymphoma cells detected with BMS/BMB were confirmed as BMinv. The morphology of lymphoma cells is characterized by a diffuse proliferation of large lymphoid cells, and its immunohistochemistry is characterized as CD20+, CD3−, CD45+, CD79a+, cyclin D1−.

## Follow-up

Follow-up was performed telephonically. Overall survival (OS) was measured from first diagnosis to the patient's death or the last follow-up date, as of 5 June 2020. Progression-free survival (PFS) was measured from first diagnosis to confirmed relapse or the last follow-up date. All patients were followed up for ≥6 months. The median PFS and OS were 21.4 and 25.2 months, respectively.

## Statistical analysis

As mentioned above, we screened patients from a convenience sample to arrive at the sample size of 171. At present, a combination of BMB/BMS and PET/CT is the standard method to evaluate BMinv [15]. Therefore, we regarded positive results of either BMB/BMS or whole-body PET/CT as confirmation of BMinv. Using these results as reference standards, we calculated the sensitivity, specificity, accuracy, positive predictive value, and negative predictive value of pelvic MRI for diagnosing BMinv in DLBCL patients. The data analysis was conducted using the R (version 3.6.0; R Foundation for Statistical Computing, Vienna, Austria) and RStudio (version 1.2.1335; RStudio, PBC, Boston, MA) software. The receiver operating characteristic (ROC) curve was constructed according to the above-mentioned sensitivity and specificity, using the "pROC" R package [16]. The area under the ROC curve (AUC) is an important indicator for evaluating diagnostic utility. The closer it is to 1.0, the higher the accuracy of the detection method; when it is equal to 0.5, the detection method has no apparent accuracy. The R package "MatchIt" [17] was used for propensity-score matching (PSM) of patients in one of six groups (BMB/BMS-positive or negative, pelvic MRI-positive or negative, and PET/CT-positive or negative), based on their clinical information (age, performance status [18], Ann Arbor stage, lactate dehydrogenase). In order to test the predictive value of each examination method on patient survival, the PFS and OS of the matched data were compared using Kaplan-Meier analysis. Chi-square or Fisher exact tests were used to compare detection rates between the two methods. All $P$ values were two-tailed, with a $P$ value of <0.05 indicating statistical significance. Further, we performed intra-class correlation (ICC) in the "irr" R package (version 0.84.1) to estimate inter-method agreement. The limits of agreement were defined as the mean ± 1.96 standard deviations of the difference between a pair of ratings. An ICC value <0.4 was defined as a poor agreement and an ICC value >0.75 was defined as a good agreement.

## Results

### Clinical characteristics of patients

In total, 171 patients (111 male and 60 female) were enrolled (Fig 1). The median age was 64 years (range: 18–78 years). Clinical characteristics are presented in Table 2.

### BMinv detection rate and consistency of BMB/BMS, PET/CT, and pelvic MRI

Among the 171 patients with DLBCL, 25 cases (14.6%) of BMinv were detected using BMB/BMS, 44 cases (25.7%) were detected using whole-body PET/CT, and 42 cases (24.6%)

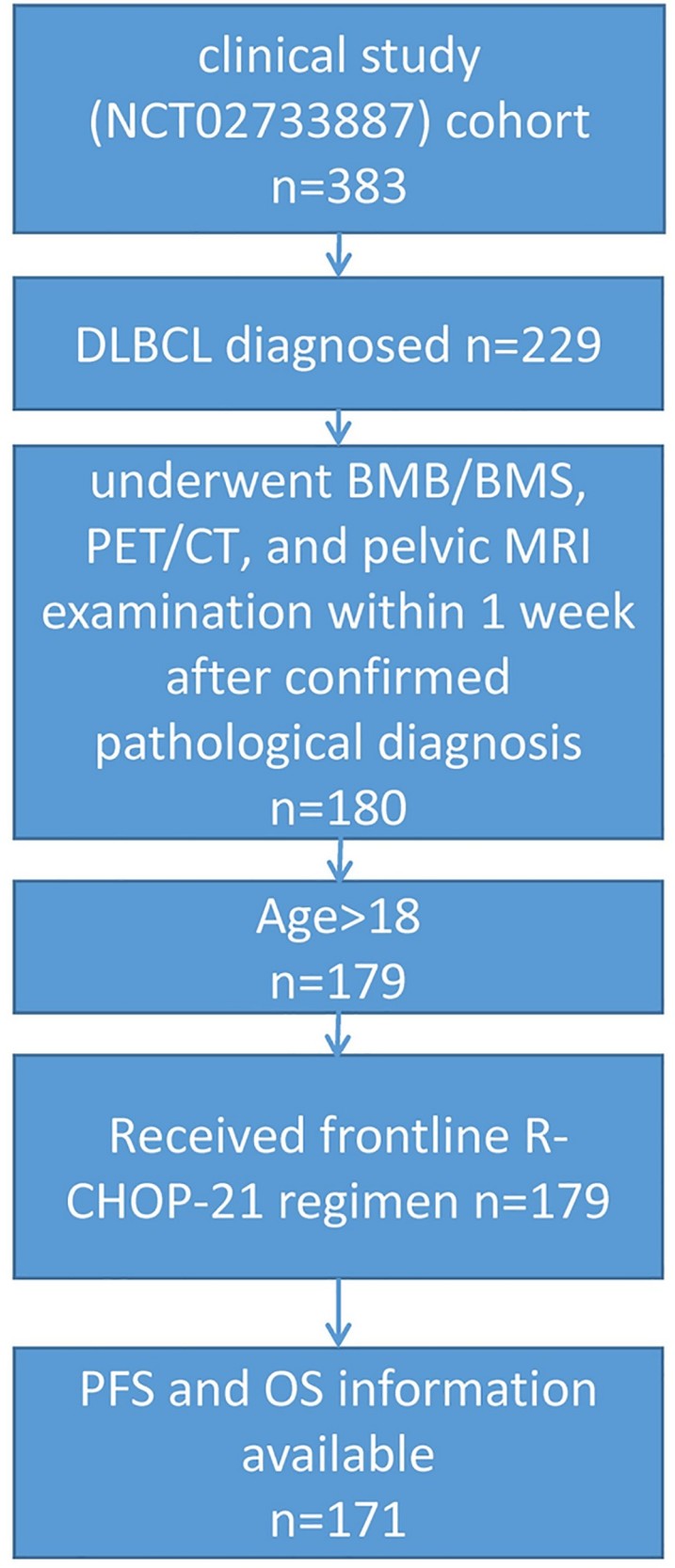

**Fig 1. Diagram of the flow of patients.** Abbreviations: DLBCL, diffuse large B-cell lymphoma; BMB/BMS, bone marrow biopsy/smear; PET/CT, positron emission tomography-computed tomography; MRI, magnetic resonance imaging; PFS, progression-free survival; OS, overall survival.

were detected using pelvic MRI. The BMinv detection rate was higher with pelvic MRI than with BMB/BMS ($\chi^2$ = 4.752, $P$ = 0.029). There was no statistically significant difference in the BMinv detection rate between pelvic MRI and PET/CT ($\chi^2$ = 0.016, $P$ = 0.901). In terms of the consistency of diagnosis using the three methods, Table 3 provides a detailed overview of ICCs.

## Diagnostic efficacy of pelvic MRI for BMinv in DLBCL

Pelvic MRI results are summarized in Table 4. The sensitivity, specificity, accuracy, positive predictive value, and negative predictive value of pelvic MRI for diagnosing BMinv in DLBCL patients were 83.33%, 98.37%, 94.15%, 95.24%, and 93.80%, respectively. The ROC curve is depicted in Fig 2. The AUC value was 0.909 (95% confidence interval: 0.854–0.963).

In order to analyze the detection performance of pelvic MRI for different subtypes of BMinv, we analyzed all pelvic MRI-positive cases in this study. Of the 42 pelvic MRI-positive cases, 19 were also BMB/BMS-positive. As presented in Table 5, only a small proportion (2/16) of pelvic MRI-positive, focal bone marrow lesions were detected using BMB/BMS. Among the pelvic MRI-positive results, 16/42 (38.1%) were of focal lesions. Among pelvic MRI-positive results, only 2/19 (focal/[diffuse + focal] = 10.5%) BMB/BMS-positive results were of focal lesions. This difference in the rate of detection of focal lesions was statistically significant (Fisher exact test: $P$ = 0.036).

## BMinv characteristics on pelvic MRI and PET/CT images

Pelvic MRI detected 42 cases with BMinv, with lesions distributed in the ilium, femur, sacro-coccyx, lumbar vertebrae, pubis, and ischium. Whole-body PET/CT detected 44 cases with BMinv, with lesions mostly distributed in the positions mentioned above. In total, 39 cases tested positive for BMinv with both techniques (PET/CT and pelvic MRI). The signal

**Table 2. Characteristics of the 171 DLBCL patients.**

| Clinical characteristics | Number of cases | Percentage |
|---|---|---|
| **Sex** | | |
| **Male** | 111 | 64.9 |
| **Female** | 60 | 35.1 |
| **Age** | | |
| **≤60 years** | 70 | 40.9 |
| **>60 years** | 101 | 59.1 |
| **Performance status** | | |
| **0 or 1** | 134 | 78.4 |
| **>1** | 37 | 21.6 |
| **Ann Arbor stage** | | |
| **I–II** | 81 | 47.4 |
| **III–IV** | 90 | 52.6 |
| **LDH (reference range: ≤285 U/L)** | | |
| **normal** | 59 | 34.5 |
| **>normal level** | 112 | 65.5 |

**Abbreviations:** DLBCL, diffuse large B-cell lymphoma; LDH, lactate dehydrogenase.

**Table 3. Intermethod agreement.**

| Comparison | ICC [95% CI] |
|---|---|
| BMB/BMS vs. PET/CT vs pelvic MRI | 0.866 [0.832, 0.895] |
| BMB/BMS vs. PET/CT | 0.864 [0.821, 0.898] |
| BMB/BMS vs. pelvic MRI | 0.829 [0.775, 0.870] |
| PET/CT vs. pelvic MRI | 0.906 [0.875, 0.929] |

**Abbreviations:** ICC, intraclass correlation; CI, confidence interval; BMB/BMS, bone marrow biopsy/bone marrow smear; PET/CT, positron emission tomography-computed tomography; MRI, magnetic resonance imaging.

representing bone marrow abnormality in the pelvic MRI scan field corresponded to the area of focal FDG uptake on the PET/CT image (Table 6), with a diffuse distribution in most cases. Two representative cases of BMinv identified using PET/CT and pelvic MRI are presented in Figs 3 and 4. PET/CT revealed an increased diffuse bone marrow FDG uptake throughout the body (higher than that in the liver) in two cases, one of which showed negative findings in pelvic MRI, BMB, and BMS. The other case showed positive findings in pelvic MRI and BMB (Figs 5 and 6).

## Propensity-score matching

In order to clarify the prognostic impact of BMinv diagnosis using BMB/BMS, pelvic MRI, and whole-body PET/CT, BMinv-positive and negative patients were matched with propensity scoring for each diagnostic method. The clinical baseline data of each of the six groups, before and after PSM, are summarized in Table 7. The distributions of propensity scores before and after matching are depicted in S1–S3 Figs. After PSM, in terms of BMB/BMS, the median PFS was 17.8 months in the BMinv-positive and 26.9 months in the BMinv-negative group ($P = 0.092$, Fig 7A), whereas the median OS was 22.3 months in the BMinv-positive and 29.8 months in the BMinv-negative group ($P = 0.240$, Fig 7B). In terms of PET/CT, the median PFS was 24.8 months in the BMinv-positive and 33.0 months in the BMinv-negative group ($P = 0.086$, Fig 8A), whereas the median OS was 27.9 months in the BMinv-positive and 33.9 months in the BMinv-negative group ($P = 0.365$, Fig 8B). In terms of pelvic MRI, the median PFS was 24.9 months in the BMinv-positive and 33.1 months in the BMinv-negative group ($P<0.001$, Fig 9A), whereas the median OS was 27.3 months in the BMinv-positive and 35.8 months in the BMinv-negative group ($P = 0.062$, Fig 9B).

## Discussion

In this study, to our knowledge, it was the first time that pelvic MRI was adopted for diagnosis of BMinv. Its beneficial characteristics include its simplicity, consistency, and efficacy. As our study contained internal controls (BMB/BMS, pelvic MRI, and whole-body PET/CT), we

**Table 4. Pelvic MRI for the diagnosis of BMinv in DLBCL (cases).**

| BMB/BMS or PET/CT | Total number of cases | Pelvic MRI | |
|---|---|---|---|
| | | Positive | Negative |
| Positive | 48 | 40 | 8 |
| Negative | 123 | 2 | 121 |

**Abbreviations:** PET/CT, positron emission tomography-computed tomography; MRI, magnetic resonance imaging; BMinv, bone marrow involvement; DLBCL, diffuse large B-cell lymphoma; BMB/BMS, bone marrow biopsy/smear.

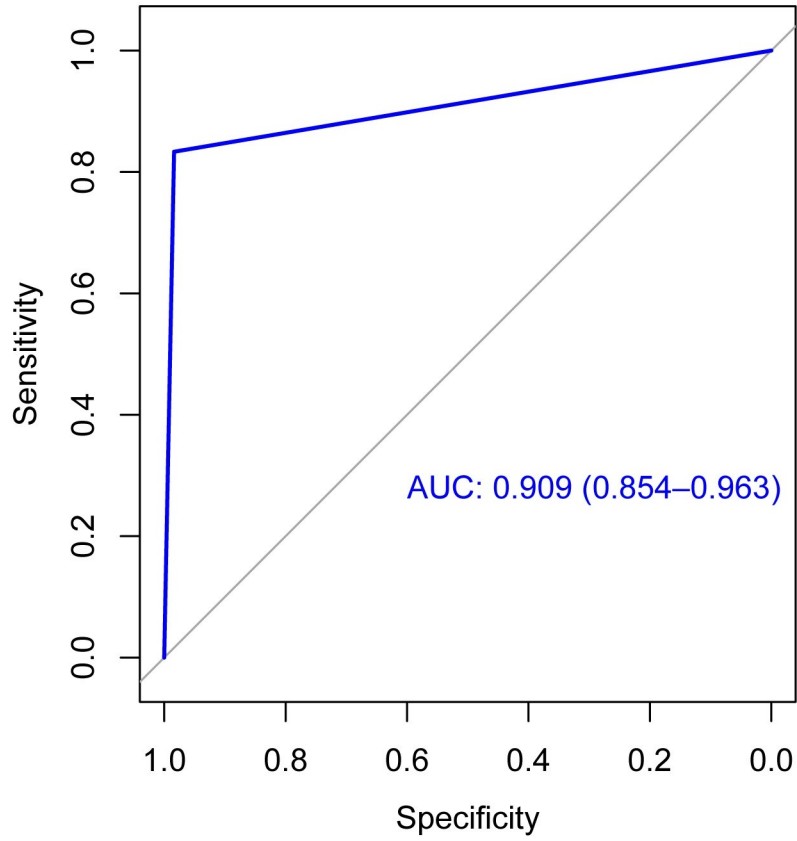

**Fig 2. Receiver operating characteristic curves.** Receiver operating characteristic curve for pelvic magnetic resonance imaging (MRI) for diagnosing bone marrow involvement in patients with diffuse large B-cell lymphoma.

believe that our results are robust. Long-term survival data is required to verify the reliability of pelvic MRI for diagnosis of BMinv.

BMinv is a common occurrence in lymphoma. A recent study [19] revealed that patients with BMinv in DLBCL had a higher risk of central nervous system recurrence and a lower progression-free survival; therefore, timely and accurate diagnosis of lymphoma BMinv is important for disease staging, choosing treatment options, and evaluating the prognosis. Current methods for clinical evaluation of BMinv in lymphoma include BMB, PET/CT, and whole-body MRI [11, 20].

BMB is the conventional method for detecting BMinv [21]. As this method is invasive, typically only the anterior superior iliac spine or posterior superior iliac spine is biopsied with one to two puncture points. However, BMinv in lymphoma is often a focal lesion, and if the puncture site is not invaded by lymphoma cells, it will not be detected. The routine bone marrow

**Table 5. Pelvic MRI-positive focal and diffuse bone lesions stratified by BMB/BMS results.**

|  | Focal bone lesions (cases) | | Diffuse bone lesions (cases) | |
| --- | --- | --- | --- | --- |
| **Pelvic MRI Positive** | 16 | | 26 | |
| **BMB/BMS** | **BMB/BMS (+)** | **BMB/BMS (-)** | **BMB/BMS (+)** | **BMB/BMS (-)** |
|  | 2 | 14 | 17 | 9 |

**Abbreviations:** MRI, magnetic resonance imaging; BMB/BMS, bone marrow biopsy/smear.

**Table 6. BMinv detected using both pelvic MRI and PET/CT in 39 DLBCL patients.**

| Examination methods | Bone marrow involvement (number of cases) | | | | | |
|---|---|---|---|---|---|---|
| | Ilium | Femur | Sacrococcyx | Lumbar vertebrae | Pubis | Ischium |
| **Pelvic MRI** | 30 | 12 | 5 | 4 | 1 | 1 |
| **PET/CT** | 31 | 11 | 6 | 5 | 2 | 1 |

**Abbreviations:** BMinv, bone marrow involvement; DLBCL, diffuse large B-cell lymphoma; MRI, magnetic resonance imaging; PET/CT, positron emission tomography-computed tomography.

puncture points do not reflect the status of the whole body; therefore, even with negative results, the possibility of BMinv cannot be completely excluded in clinical practice. In our study, only 25 patients with DLBCL were diagnosed with BMinv using BMB/BMS.

PET/CT is a technique that can realize the fusion of functional and structural images, and can be used to distinguish tumor cells from normal cells according to their differing FDG uptake [22–23]. This technique is widely recommended for diagnosis and staging of lymphoma, as well as for the therapeutic evaluation of lymphoma. The efficacy of PET/CT for the evaluation of lymphoma BMinv has been analyzed [10–11, 24]. It was discovered that, when BMB results were negative and PET/CT results were positive, the false negative rate could be reduced using repeated/multiple BMB guided with positive lesions on PET/CT [25–27]. The sensitivity of PET/CT in the diagnosis of different subtypes of lymphoma differs greatly; in DLBCL, it exhibits high sensitivity [23]. At present, various guidelines [6, 15] recommend PET/CT and/or BMB as standard methods to evaluate BMinv in DLBCL. If PET/CT is positive, BMinv can be diagnosed without BMB. However, PET/CT is quite expensive, which limits its application in the field of medicine [28]. Moreover, PET/CT exposes patients to high doses of radiation, especially as it may be used repeatedly, such as for staging and post-treatment evaluation. However, digital PET/CT may improve this situation. In a study by Sekine et al [29], clinically sufficient PET image quality was obtained with simulated reduction of up to 40% of the standard injected dose of FDG when using silicon photomultiplier (SiPM) detectors. Thus, substantially lower radiation doses can be achieved with PET imaging using SiPM detectors compared with those in conventional PET/CT. In addition, simultaneous PET/MRI

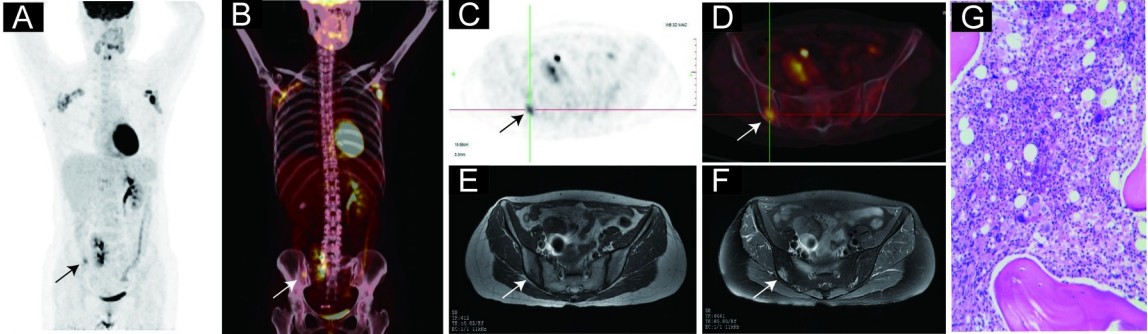

**Fig 3. Representative case of a 46-year-old female diagnosed with diffuse large B-cell lymphoma.** (A) and (B): Coronal positron emission tomography (PET) and PET-computed tomography (CT) images, respectively, revealing a lesion with an increased focal [18]F-fluorodeoxyglucose uptake in the right posterior superior iliac spine. (C) and (D): Transverse view of the same process as in panels A and B, respectively. (E): The right posterior superior iliac spine exhibiting low signal intensity on the T1-weighted, transverse pelvic magnetic resonance image (MRI). (F): The right posterior superior iliac spine exhibiting high signal intensity on the fat-suppressed, T2-weighted, transverse pelvic MRI. (G): Hematoxylin and eosin staining confirmed bone marrow involvement in this patient. The patient's blood glucose level before the PET/CT imaging was 4.9 mmol/L.

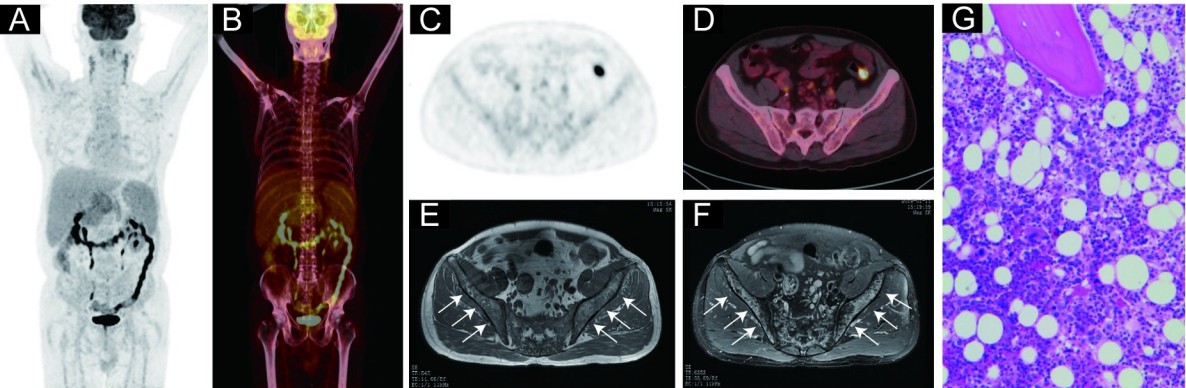

**Fig 4. Representative case of a 55-year-old male diagnosed with diffuse large B-cell lymphoma.** (A) and (B): Coronal positron emission tomography (PET) and PET-computed tomography (CT) images, respectively, revealing no lesions with increased focal [18]F-fluorodeoxyglucose (FDG) uptake (higher than that in the liver) in any bone marrow throughout the patient's body. (C) and (D): Transverse PET and PET/CT images, respectively, revealing no lesions with increased focal FDG uptake in the right and left ilium. (E): The right and left ilium exhibiting multiple areas of low signal intensity on the T1-weighted, transverse pelvic magnetic resonance image (MRI). (F): The right and left ilium exhibiting multiple areas of high signal intensity on the fat-suppressed, T2-weighted, transverse pelvic MRI. (G): Hematoxylin and eosin staining confirmed bone marrow involvement in this patient. The patient's blood glucose level before the PET/CT imaging was 7.7 mmol/L.

combines the metabolic information obtained using PET with the superior soft-tissue resolution obtained using MRI. Previous studies have demonstrated that simultaneous PET/MRI is clinically feasible in patients with lymphoma [30, 31]. PET/MRI performance was demonstrated to be comparable to that of PET/CT for lesion detection and measurements of

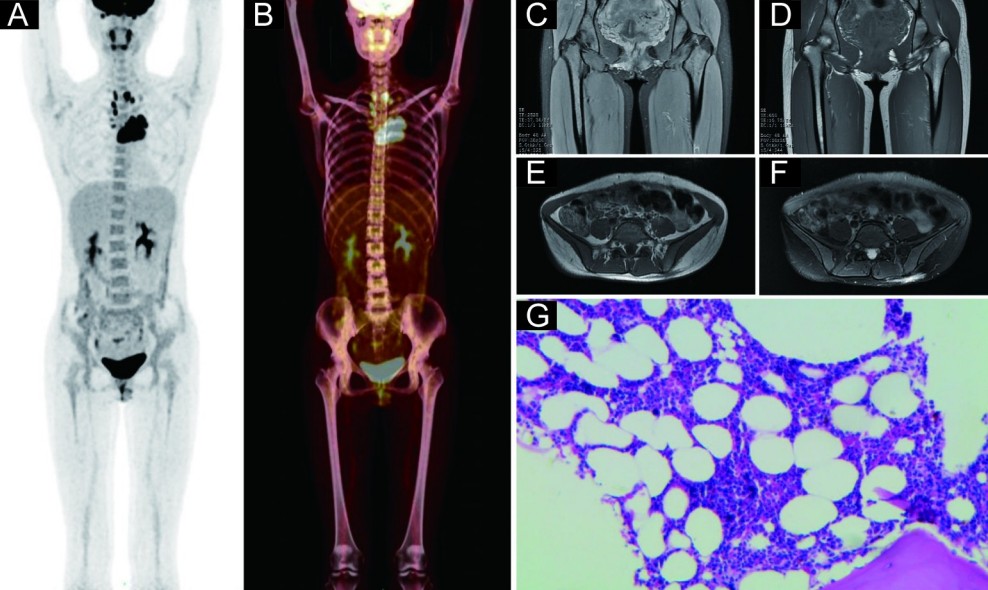

**Fig 5. Representative case of a 30-year-old female diagnosed with diffuse large B-cell lymphoma.** (A) and (B): Coronal positron emission tomography (PET) and PET-computed tomography images, respectively, revealing increased diffuse [18]F-fluorodeoxyglucose uptake (higher than that in the liver) in bone marrow throughout the patient's body. (C)–(F): Pelvic magnetic resonance imaging revealing no abnormal signals on the T1-weighted or fat-suppressed T2-weighted images. (G): Hematoxylin and eosin staining confirmed that there was no bone marrow involvement in this patient. The patient's blood glucose level before the PET/CT imaging was 4.1 mmol/L.

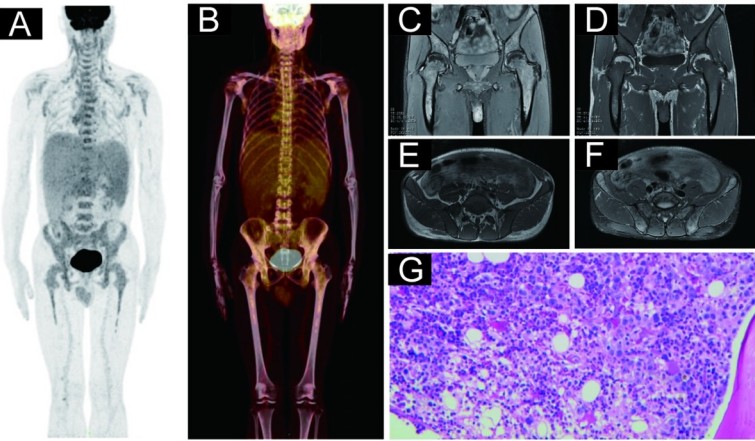

**Fig 6. Representative case of a 45-year-old male diagnosed with diffuse large B-cell lymphoma.** (A) and (B): Coronal positron emission tomography (PET) and PET-computed tomography images, respectively, revealing increased diffuse $^{18}$F-fluorodeoxyglucose uptake (higher than that in the liver) in bone marrow throughout the patient's body. (C)–(F): Pelvic magnetic resonance imaging revealing low signal intensity in the ilium, bilateral upper femur bones, and sacrum on the T1-weighted image, and high signal intensity in the same regions on the fat-suppressed T2-weighted image. (G): Hematoxylin and eosin staining confirmed bone marrow involvement in this patient. The patient's blood glucose level before the PET/CT imaging was 5.0 mmol/L.

standardized uptake values. Replacement of PET/CT with PET/MRI would markedly decrease the radiation doses to which lymphoma patients are exposed during diagnostic imaging.

MRI is an imaging method that does not make use of ionizing radiation. In recent years, it has been confirmed that BMB has a low sensitivity in the diagnosis of lymphoma BMinv, and that whole-body MRI can improve this detection rate [11, 20]. There are also several studies demonstrating that whole-body MRI has a similar sensitivity and specificity to that of PET/CT [9–11, 32]. However, there are several challenges with whole-body MRI that are hard to overcome. Its resolution is lower than that of pelvic MRI; it is subject to motion artifacts, complicating its use in diagnosis of BMinv of the thoracic spine and ribs, due to patient breathing; and it is expensive, time-consuming, and not available in all hospitals, which limits its clinical application. Lymphoma is a systemic disease and, according to present guidelines, if any BMinv is detected, it can be classified as stage IV without the necessity to identify all bone marrow lesions. It has been reported that the most common sites of bone marrow lesions associated with lymphoma in adults are the vertebrae, pelvis, proximal femoral metaphyses, and skull, with incidence rates of 69%, 41%, 25%, and 14%, respectively [12]. Pelvic MRI can simultaneously reveal lymphoma bone marrow lesions of the pelvis, the femur, and parts of the vertebral body, and its examination range is far beyond the bone marrow puncture point. In our study, 25 cases of BMinv were confirmed using BMB/BMS. The BMinv detection rate of pelvic MRI was higher than that of BMB/BMS; as expected, a large proportion of pelvic MRI-positive focal bone marrow lesions were not detected using BMB/BMS. However, there was no difference between the BMinv detection rate of pelvic MRI and that of PET/CT. Upon ROC-curve analysis, we determined that the AUC value for pelvic MRI was close to 1.0, with satisfactory sensitivity and specificity values, indicating that it has high diagnostic value to identify BMinv in DLBCL. The consistency between the three methods was good (ICC>0.75), with PET/CT and pelvic MRI exhibiting the highest diagnostic agreement. Therefore, our pelvic MRI procedure may be a viable alternative to other BMinv imaging methods.

BMinv in lymphoma can be focal or diffuse [11, 14]. In the present study, we compared the distribution characteristics of the lesions detected using the two techniques, and discovered

**Table 7. Patient characteristics for different groups before and after propensity score matching.**

| Stratification factor | Before propensity score matching | | | | | | | | |
|---|---|---|---|---|---|---|---|---|---|
| | BMB/BMS | | | PET/CT | | | Pelvic MRI | | |
| | BMinv-negative (N = 146) | BMinv-positive (N = 25) | p value | BMinv-negative (N = 127) | BMinv-positive (N = 44) | p value | BMinv-negative (N = 129) | BMinv-positive (N = 42) | p value |
| Age | | | 0.060 | | | 0.008 | | | 0 |
| ≤60 years | 55 (37.7%) | 15 (60.0%) | | 44 (34.6%) | 26 (59.1%) | | 46 (35.7%) | 24 (57.1%) | |
| >60 years | 91 (62.3%) | 10 (40.0%) | | 83 (65.4%) | 18 (40.9%) | | 83 (64.3%) | 18 (42.9%) | |
| Ann Arbor staging | | | 0.059 | | | 0 | | | 0 |
| I-II | 74 (50.7%) | 7 (28.0%) | | 73 (57.5%) | 8 (18.2%) | | 72 (55.8%) | 9 (21.4%) | |
| III-IV | 72 (49.3%) | 18 (72.0%) | | 54 (42.5%) | 36 (81.8%) | | 57 (44.2%) | 33 (78.6%) | |
| Performance status | | | 0.270 | | | 0.003 | | | 0.020 |
| 0 or 1 | 117 (80.1%) | 17 (68.0%) | | 107 (84.3%) | 27 (61.4%) | | 107 (82.9%) | 27 (64.3%) | |
| >1 | 29 (19.9%) | 8 (32.0%) | | 20 (15.7%) | 17 (38.6%) | | 22 (17.1%) | 15 (35.7%) | |
| LDH | | | 1 | | | 0.085 | | | 0.136 |
| normal | 50 (34.2%) | 9 (36.0%) | | 49 (38.6%) | 10 (22.7%) | | 49 (38.0%) | 10 (23.8%) | |
| >normal level | 96 (65.8%) | 16 (64.0%) | | 78 (61.4%) | 34 (77.3%) | | 80 (62.0%) | 32 (76.2%) | |
| Stratification factor | After propensity score matching | | | | | | | | |
| | BMB/BMS | | | PET/CT | | | Pelvic MRI | | |
| | BMinv-negative (N = 25) | BMinv-positive (N = 25) | p value | BMinv-negative (N = 44) | BMinv-positive (N = 44) | p value | BMinv-negative (N = 42) | BMinv-positive (N = 42) | p value |
| Age | | | 1 | | | 0.521 | | | 1 |
| ≤60 years | 15 (60.0%) | 15 (60.0%) | | 22 (50.0%) | 26 (59.1%) | | 24 (57.1%) | 24 (57.1%) | |
| >60 years | 10 (40.0%) | 10 (40.0%) | | 22 (50.0%) | 18 (40.9%) | | 18 (42.9%) | 18 (42.9%) | |
| Ann Arbor staging | | | 1 | | | 1 | | | 1 |
| I-II | 7 (28.0%) | 7 (28.0%) | | 7 (15.9%) | 8 (18.2%) | | 8 (19.0%) | 9 (21.4%) | |
| III-IV | 18 (72.0%) | 18 (72.0%) | | 37 (84.1%) | 36 (81.8%) | | 34 (81.0%) | 33 (78.6%) | |
| Performance status | | | 1 | | | 0.825 | | | 1 |
| 0 or 1 | 17 (68.0%) | 17 (68.0%) | | 29 (65.9%) | 27 (61.4%) | | 29 (69.0%) | 27 (64.3%) | |
| >1 | 8 (32.0%) | 8 (32.0%) | | 15 (34.1%) | 17 (38.6%) | | 13 (31.0%) | 15 (35.7%) | |
| LDH | | | 1 | | | 0.628 | | | 0.625 |
| normal | 9 (36.0%) | 9 (36.0%) | | 13 (29.5%) | 10 (22.7%) | | 13 (31.0%) | 10 (23.8%) | |
| >normal level | 16 (64.0%) | 16 (64.0%) | | 31 (70.5%) | 34 (77.3%) | | 29 (69.0%) | 32 (76.2%) | |

**Abbreviations:** BMinv, bone marrow involvement; BMB, bone marrow biopsy; BMS, bone marrow smear; PET, positron emission tomography; CT, computed tomography; MRI, magnetic resonance imaging; LDH, lactate dehydrogenase.

that the abnormal signal on pelvic MRI and the focal increased FDG uptake on PET/CT mostly overlapped. In most cases, the BMinv was diffuse. Lesions detected using pelvic MRI were most prevalent in the ilium and femur, while those detected using PET/CT were most prevalent in the pelvis, femur, and vertebral body. Of the 44 patients with positive whole-body PET/CT findings for BMinv, 41 revealed at least one bone marrow lesion in the ilium, lumbar vertebrae, sacrococcyx, ischium, pubis, and upper femur; these findings were generally consistent with the distribution of pelvic lesions on MRI. This is also the reason why pelvic MRI and PET/CT exhibited similar BMinv detection rates. These findings indicate that pelvic MRI can be used to diagnose most instances of BMinv in DLBCL patients, and its ability to detect BMinv in DLBCL is comparable to that of PET/CT.

In our study, PET/CT indicated increased diffuse bone marrow FDG uptake throughout the body (higher than that in the liver) in two cases. One of these cases indicated negative

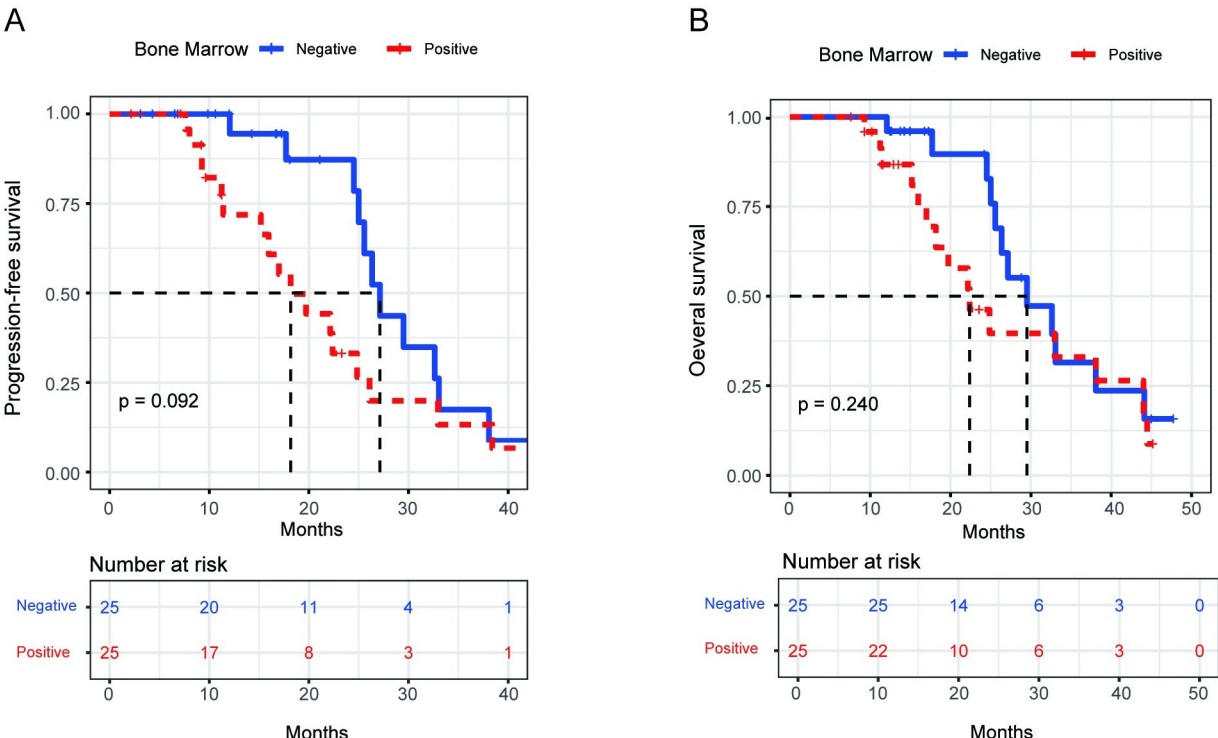

**Fig 7. Survival curves of patients diagnosed using bone marrow biopsy/smear after propensity-score matching.** (A): Progression-free survival curve. (B): Overall survival curve.

findings in pelvic MRI, BMB, and BMS. This patient had symptoms of fever, and increased diffuse FDG uptake due to reactive bone marrow hyperplasia or inflammatory bone marrow lesions could not be ruled out. Thus, pelvic MRI may be more advantageous in such patients. In the other case, BMB of the posterior superior iliac spine indicated positive findings, and pelvic MRI indicated abnormal signal changes in the ilium, bilateral upper femur bones, and sacrum. The results of these cases indicate that pelvic MRI can help in diagnosing BMinv in patients with increased diffuse FDG uptake in bone marrow; this should be confirmed in future studies with more cases. However, it remains unclear whether diffuse FDG uptake in bone marrow on PET/CT should be considered BMinv in DLBCL [10, 15, 24].

In a meta-analysis of seven clinical studies including a total of 654 DLBCL patients, conducted by Adams et al [10], four studies showed that diffuse FDG uptake in bone marrow indicated positive BMinv. Moreover, of 14 patients with increased diffuse FDG uptake in the bone marrow on PET/CT, 12 were confirmed to have BMinv using BMB. Khan et al [24] conducted PET/CT examinations for 130 DLBCL patients and found that 33 showed positive findings for BMinv on PET/CT; of these, 28, 3, and 2 cases showed focal FDG uptake, focal FDG uptake with increased diffuse FDG uptake, and increased diffuse FDG uptake, respectively. The two patients with diffuse FDG uptake underwent BMB, which confirmed the presence of BMinv. Current European Society for Medical Oncology clinical practice guidelines for DLBCL use focal bone marrow FDG uptake with or without increased diffuse FDG uptake (higher than that in the liver) upon PET/CT, as a criterion for the diagnosis of BMinv [15]. There are many reasons for increased diffuse FDG uptake in bone marrow, including reactive hyperplasia, inflammatory myelopathy, and tumor invasion. Although some patients with increased diffuse

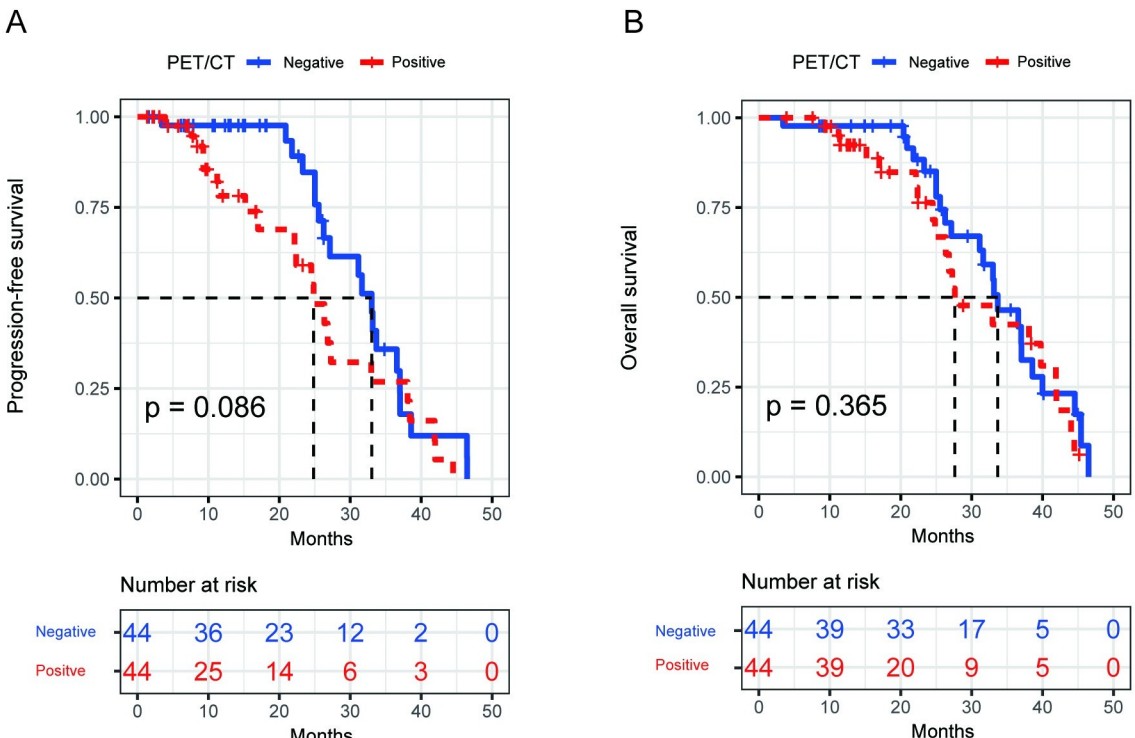

**Fig 8. Survival curves of patients diagnosed using positron emission tomography-computed tomography after propensity-score matching.** (A): Progression-free survival curve. (B): Overall survival curve.

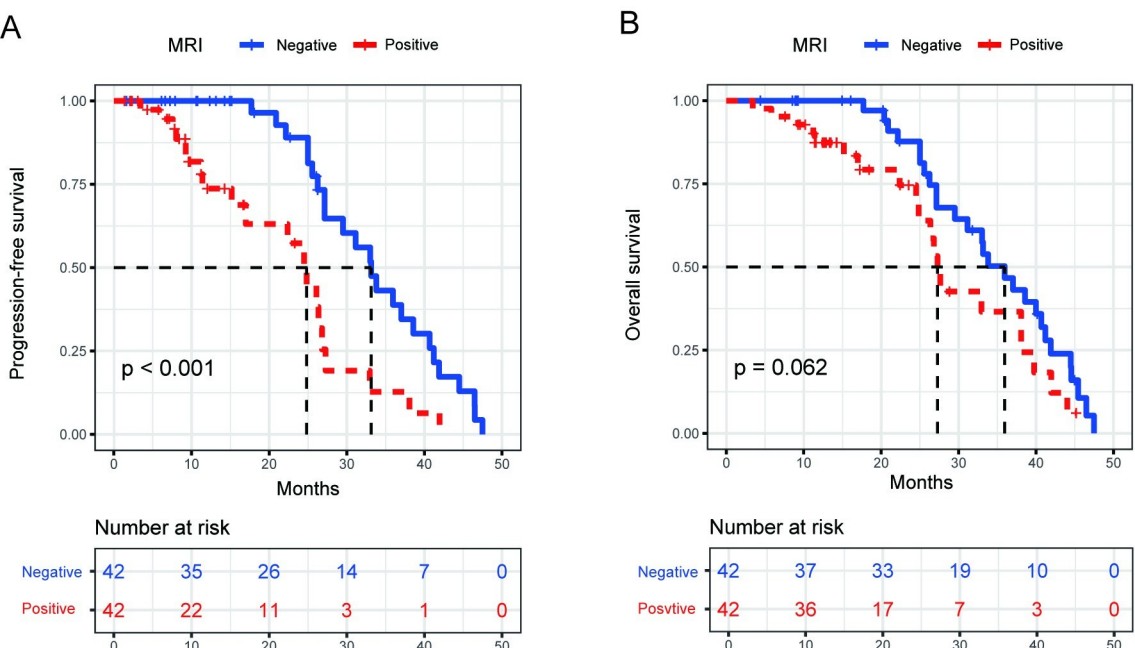

**Fig 9. Survival curves of patients diagnosed using pelvic magnetic resonance imaging after propensity score matching.** (A): Progression-free survival curve. (B): Overall survival curve.

bone marrow FDG uptake have been confirmed to have BMinv after BMB, the diagnostic value of these imaging findings for BMinv in lymphoma requires further research. In our study, three patients exhibited positive findings for BMinv in both BMB/BMS and PET/CT, whereas pelvic MRI was negative for these patients. In such cases, pelvic MRI examination alone is insufficient.

One limitation of this study was that we did not evaluate the reproducibility of our pelvic MRI protocol for diagnosing BMinv on 1.5T MRI scanners. In previous studies, it was demonstrated that multiparametric MRI modules, such as iron-corrected T1 and T2*, were repeatable and reproducible across different scanner manufacturers (Philips and Siemens) and field strengths (1.5T and 3.0T) [33, 34]. They also demonstrated that a large difference in fat content between tissues affects the reproducibility of those MRI modalities. Future studies will be needed to confirm whether 1.5T MRI scanners can be used for accurate pelvic MRI diagnosis of BMinv.

Further, our study had a retrospective design. Prospective studies with larger sample sizes and with long follow-up periods may help to illustrate the diagnostic value of pelvic MRI in terms of BMinv in patients with DLBCL. Another limitation was in our protocol for pelvic MRI, which may have increased the false positive rate using this technique. When BMB is performed before MRI on patients with lymphoma, it may result in an area of bone edema in the posterior superior iliac crest that is hyperintense on T2 images and may be incorrectly interpreted as a lesion [35]. Therefore, when possible, pelvic MRI should be performed before BMB. If BMB is performed first, there are two solutions. The first is to exclude the BMB site from the MRI evaluation. The second is to perform multi-b-value diffusion-weighted imaging to improve the signal-to-noise ratio [36, 37]. This aspect should be taken into consideration in future studies.

In addition, we tested the reliability and practicability of the three diagnostic methods by following up patient survival rates after PSM to balance baseline clinical data between groups [38] and minimize the bias it can cause. From our study, pelvic MRI seems to be better able to distinguish PFS than both BMB/BMS and PET/CT are ($P<0.001$). For patients diagnosed with BMinv using pelvic MRI, the necessity to increase the intensity of chemotherapy or shorten the interval between follow-ups warrants further research. In any case, MRI is advantageous in its use of non-ionizing radiation, its non-invasive nature, and in its independence of intravenous contrast agents [39, 40].

The reference standard we have used combines classic BMB/BMS with PET/CT, as recommended in the current diagnosis and treatment guidelines, to reduce the possibility of false negative tests. In any case, survival analysis performed in this study demonstrated the efficacy of pelvic MRI in comparison to the reference standards.

## Conclusion

The false negative rate for BMinv in DLBCL is higher with BMB/BMS than with pelvic MRI and PET/CT, and the latter two techniques have a high diagnostic value for BMinv. Furthermore, pelvic MRI, as diagnostic tool for diagnosis of BMinv, has a higher predictive value of progression-free survival of patients than BMB/BMS and PET/CT has. At present, a combination of BMB/BMS and PET/CT is the standard method to evaluate BMinv. However, few hospitals in China are currently equipped with PET/CT instruments, and examination costs are high. For patients who are unable to undergo PET/CT, particularly in economically underdeveloped areas, pelvic MRI may not only be an alternative, but also actually improve the BMinv detection rate for DLBCL patients. Thus, pelvic MRI may ultimately be used to improve the accuracy of clinical staging, guide patient treatment, and evaluate prognosis.

## Supporting information

**S1 Fig. Distribution of propensity scores of BMB/BMS results.**
(TIF)

**S2 Fig. Distribution of propensity scores of PET/CT results.**
(TIF)

**S3 Fig. Distribution of propensity scores of pelvic MRI results.**
(TIF)

**S1 Data. Original data for the present study is available online.**
(CSV)

## Acknowledgments

The authors are grateful to the diagnosticians who assisted in this study, namely Dr. Ling-Sha Huang, Dr. Mei-Qi Li, Dr. Guo-You Xiao, Dr. Xin Zhao, Dr. Dong Xie, and Dr. Zheng Wang, as well as the patient representatives at Guangxi Medical University Cancer Hospital, Nanning 530021, P.R. China, namely Dan Ping Fu and Lin Kai Liang.

## Author Contributions

**Conceptualization:** Qing Ke, Cheng-Cheng Liao.

**Data curation:** Qing Ke, Cheng-Cheng Liao, Xiao-Hong Tan.

**Formal analysis:** Qing Ke.

**Funding acquisition:** Hong Cen, Le-Qun Li.

**Investigation:** Xiao-Hong Tan.

**Project administration:** Qing Ke, Xiao-Hong Tan, Hong Cen, Le-Qun Li.

**Resources:** Cheng-Cheng Liao.

**Software:** Cheng-Cheng Liao.

**Supervision:** Cheng-Cheng Liao, Xiao-Hong Tan, Bao-Ping Guo, Hong Cen, Le-Qun Li.

**Validation:** Bao-Ping Guo.

**Visualization:** Bao-Ping Guo.

**Writing – original draft:** Qing Ke, Cheng-Cheng Liao.

**Writing – review & editing:** Qing Ke.

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
