## [Decision Letter · Decision Letter 0]

20 Jan 2021

PONE-D-20-35498

Diagnostic accuracy of pelvic magnetic resonance imaging for the assessment of bone marrow involvement in diffuse large B-cell lymphoma

PLOS ONE

Dear Dr. Li,

Thank you for submitting your manuscript to PLOS ONE. After careful consideration, we feel that it has merit but does not fully meet PLOS ONE’s publication criteria as it currently stands. Therefore, we invite you to submit a revised version of the manuscript that addresses the points raised during the review process.

We look forward to receiving your revised manuscript.

Kind regards,

Matteo Bauckneht

Academic Editor

PLOS ONE

Journal Requirements:

2. We noted that you refer to your original study as a clinical trial through out the manuscript, but according to your description and the WHO definition of clinical trials we would not consider this a clinical trial, since although patients were prospectively assigned to both imaging modalities, there does not appear to have been any analysis of the specific effects of the imaging modality on patient health outcomes, either directly or as a result of differential treatment based on diagnosis. In order to avoid confusion we would suggest that you change the wording in your manuscript and avoid referring to this study as a clinical trial.

"NO - The funders had no role in study design, data collection and analysis, decision to publish, or preparation of the manuscript."

Reviewers' comments:

Reviewer's Responses to Questions

**Comments to the Author**

1. Is the manuscript technically sound, and do the data support the conclusions?

Reviewer #1: Partly

Reviewer #2: Yes

2. Has the statistical analysis been performed appropriately and rigorously? 

Reviewer #1: Yes

Reviewer #2: Yes

3. Have the authors made all data underlying the findings in their manuscript fully available?

Reviewer #1: Yes

Reviewer #2: Yes

4. Is the manuscript presented in an intelligible fashion and written in standard English?

Reviewer #1: Yes

Reviewer #2: Yes

5. Review Comments to the Author

Reviewer #1: This is an interesting study on a large and homogenous sample of DLBCL patients.

Here my comments, my major concern is included in comment 10:

1. Lines 52-53, I wouldn’t say “it is not difficult to treat”, but of course I agree that it has high relapse rate and poor prognosis. Please rephrase.

2. Lines 53-54, BMB actually IS currently the main method used for clinical assessment of BMI. It is not only used to identify positive/negative BMI patients, but also to evaluate marrow cellularity, hematopoietic reserve, and some prognostic features. This should be mentioned, given that you’ve compared this procedure with two imaging modalities that cannot provide this data.

3. Lines 82-83, I wouldn’t write “patients with BMB/bone marrow smear (BMS), pelvic MRI, or PET/CT test contraindications” among the exclusion criteria. Indeed, this is not a prospective study and you are not enrolling patients, so you should include “availability of BMB, PET, and MRI” among the inclusion criteria.

4. Clinical data, didn’t you have a minimum follow-up? Reading follow-up paragraph it seems that a minimum of 7 months was considered, it should be added to your inclusion criteria.

5. Pelvic MRI, I would add the planes (both sequences on coronal and transverse plane?).

6. Pelvic MRI, Lines 101-105, if you strictly follow these criteria for image interpretation, you might wrongly define as focal bone locations signal changes related to recent BMB in the iliac crest. How did you differentiate post-procedural marrow changes from lymphomatous lesions? I suggest you underline that a MRI pitfall can be associated with BMB procedure as already reported [Pitfalls in whole body MRI with diffusion weighted imaging performed on patients with lymphoma: What radiologists should know. Magn Reson Imaging. 2016;34(7):922-31. doi: 10.1016/j.mri.2016.04.023.].

7. Pelvic MRI, didn’t you differentiate focal bone lesions from diffuse BMI? This should be better specified. Focal bone lesions of course cannot be detected by blinded BMB, this should be highlighted to partly justify data concerning the comparison BMB vs imaging tests.

8. Pelvic MRI, please add years of experience of both readers

9. PET-CT, please add years of experience of both readers

10. PET/CT, I see that you reviewed whole-body PET/CT images, but it is not clear if you considered whole-body PET/CT or only pelvic PET/CT images for the comparison with pelvic MRI. This is crucial and should be clarified. Indeed, reading the Results, in Table 4 I see only pelvic PET/CT locations. Didn’t you consider possible rib, cervico-thoracic spine or other bone locations? This is not consistent with clinical practice and shouldn’t be done in a study aimed to validate pelvic MRI as a diagnostic test in the work-up of lymphoma patients, thus you cannot simply include this point as a limitation of the study. So, even though I understand that common locations are in pelvic bones, if you considered only pelvic bone locations on PET/CT, I strongly suggest to review all PET/CT images to include other potential bone marrow locations in order to assess the actual diagnostic value of pelvic MRI. It should be investigated the clinical and prognostic impact of missing non-pelvic bone locations by using only pelvic MRI, this is essential to support the use of pelvic MRI in clinical practice. Alternatively, it should be underlined that your purpose was to assess the effectiveness of MRI and not of pelvic MRI as a specific diagnostic tool, although limiting the evaluation of MRI and PET/CT to the pelvic bones of course improve the accuracy and agreement of these modalities, reducing the risk of missing other locations (e.g. ribs can be challenging in WB-MRI).

11. BMB/BMS, please add years of experience of both pathologists. Further, you should improve this paragraph including more info about histology and immunohistochemistry analysis

12. Statistical analysis, I suggest you to assess also the agreement between the three tests, not only to compare the detection rate.

13. Limitations, Line 352, “included only patients with DLBCL”. This is not a limitation, indeed BMB is no longer used in HL, while it will remain essential in marginal lymphoma to evaluate marrow cellularity and hematopoietic reserve. The issue of BMI in especially in DLBCL in which BMI is frequent and often presents as focal rather than diffuse (as in marginal zone), with still unclear role of diffuse FDG uptake on PET.

14. Lines 360-361, I would also underline that you didn’t inject contrast media during MRI, this is another strength as suggested by several authors [Whole body magnetic resonance in indolent lymphomas under watchful waiting: The time is now. Eur Radiol. 2018 Mar;28(3):1187-1193.; Whole-Body Magnetic Resonance Imaging in Oncology: Uses and Indications. Magn Reson Imaging Clin N Am. 2018 Nov;26(4):495-507].

Reviewer #2: Dear Authors,

This is an article about the diagnostic accuracy of pelvic MRI in BM involvement of DLCBL. Your efforts should be commended and the issue is of relevant interest for the Scientific Community. Also, the manuscript is well written and easily readable.

However, I have a few requests:

1) Please, in the whole manuscript use a different acronym than BMI (i.e., BMinv) because it could be misleading.

2) It would be interesting to add in the figures’ legends, the blood glucose level of each correspondent patient.

3) Page 17 line 295: “PET/CT exposes patients to high doses of radiation”.

Do you refer to PET with diagnostic CT?. Also, you cited a paper of 2014. I would like that you briefly comments about the Digital PET/CT upgrade in terms of radiation exposure (reduction of almost 20% per scan), and other technical opportunities with appropriate references.

4) Do you consider that the paper’s results are also reproducible for 1.5T MRI scanner? Please, briefly comment on this potential issue.

5) Please, also discuss the potentiality of the obvious synergies of simultaneous PET/MRI in such a scenario.

6. PLOS authors have the option to publish the peer review history of their article (what does this mean?). If published, this will include your full peer review and any attached files.

Reviewer #1: No

Reviewer #2: **Yes: **Riccardo Laudicella

---

## [Author Response · Author response to Decision Letter 0]

4 May 2021

Reviewer #1: This is an interesting study on a large and homogenous sample of DLBCL patients.

Here my comments, my major concern is included in comment 10:

1. Lines 52-53, I wouldn’t say “it is not difficult to treat”, but of course I agree that it has high relapse rate and poor prognosis. Please rephrase.

Response: Thank you for your comment. I have removed the phrase, “it is not difficult to treat.” 

Revised text, lines 51–53: “Bone marrow assessment forms part of the Ann Arbor staging system, and if a patient exhibits BMinv, the lymphoma is classified as stage IV, which has a high relapse rate and poor prognosis.”

2. Lines 53-54, BMB actually IS currently the main method used for clinical assessment of BMI. It is not only used to identify positive/negative BMI patients, but also to evaluate marrow cellularity, hematopoietic reserve, and some prognostic features. This should be mentioned, given that you’ve compared this procedure with two imaging modalities that cannot provide this data.

Response: Thank you for your comment. I have revised the text accordingly.

Revised text, lines 53–55: "Bone marrow biopsy (BMB) is the main method used for clinical assessment of BMinv. It is also used to evaluate marrow cellularity, hematopoietic reserve, and certain prognostic features.”

3. Lines 82-83, I wouldn’t write “patients with BMB/bone marrow smear (BMS), pelvic MRI, or PET/CT test contraindications” among the exclusion criteria. Indeed, this is not a prospective study and you are not enrolling patients, so you should include “availability of BMB, PET, and MRI” among the inclusion criteria.

Response: Thank you for the suggestion. I have modified this “exclusion criterion” to incorporate it as an inclusion criterion.

Revised text, lines 78–79: “(4) patients for whom BMB/bone marrow smear (BMS), pelvic MRI, and whole-body PET/CT were available;”

4. Clinical data, didn’t you have a minimum follow-up? Reading follow-up paragraph it seems that a minimum of 7 months was considered, it should be added to your inclusion criteria.

Response: Thank you for your question. The minimum follow-up was 6 months. I have added this as an inclusion criterion.

Revised text, line 79: “(5) patients followed up for ≥6 months;”

5. Pelvic MRI, I would add the planes (both sequences on coronal and transverse plane?).

Response: Thank you for the suggestion. Both sequences were indeed performed in the coronal and transverse planes. I have added this description to the text.

Revised text, lines 102–104: “Routine, T1-weighted image and fat-suppressed, T2-weighted image sequence scans were performed, each in both the coronal and transverse planes.”

6. Pelvic MRI, Lines 101-105, if you strictly follow these criteria for image interpretation, you might wrongly define as focal bone locations signal changes related to recent BMB in the iliac crest. How did you differentiate post-procedural marrow changes from lymphomatous lesions? I suggest you underline that a MRI pitfall can be associated with BMB procedure as already reported [Pitfalls in whole body MRI with diffusion weighted imaging performed on patients with lymphoma: What radiologists should know. Magn Reson Imaging. 2016;34(7):922-31. doi: 10.1016/j.mri.2016.04.023.].

Response: Thank you for your advice. It is a very important suggestion. I added this information to the discussion.

Revised text, lines 417–424: “Another limitation was in our protocol for pelvic MRI, which may have increased the false positive rate using this technique. When BMB is performed before MRI on patients with lymphoma, it may result in an area of bone edema in the posterior superior iliac crest that is hyperintense on T2 images and may be incorrectly interpreted as a lesion (Albano D, La Grutta L, Grassedonio E, Patti C, Lagalla R, Midiri M, et al. Pitfalls in whole body MRI with diffusion weighted imaging performed on patients with lymphoma: what radiologists should know. Magn Reson Imaging. 2016;34: 922–931.).Therefore, when possible, pelvic MRI should be performed before BMB. If BMB is performed first, there are two solutions. The first is to exclude the BMB site from the MRI evaluation. The second is to perform multi-b-value, diffusion-weighted imaging to improve the signal-to-noise ratio (Koh DM, Blackledge M, Padhani AR, Takahara T, Kwee TC, Leach MO, et al. Whole-body diffusion-weighted MRI: tips, tricks, and pitfalls. Am J Roentgenol. 2012;199: 252–262; Padhani AR, Liu G, Koh DM, Chenevert TL, Thoeny HC, Takahara T, et al. Diffusion-weighted magnetic resonance imaging as a cancer biomarker: consensus and recommendations. Neoplasia. 2009;11: 102–125). This aspect should be taken into consideration in future studies.”

7. Pelvic MRI, didn’t you differentiate focal bone lesions from diffuse BMI? This should be better specified. Focal bone lesions of course cannot be detected by blinded BMB, this should be highlighted to partly justify data concerning the comparison BMB vs imaging tests.

Response: Thank you for your comment. This is a good suggestion. Pelvic MRI can indeed be used to differentiate focal bone lesions from diffuse BMinv. I added relevant data to the text, comparing focal and diffuse bone lesions detected using pelvic MRI.

Revised text, lines 109–110: “The pattern of BMinv was classified as focal or diffuse.”

Revised text, lines 194–201: “In order to analyze the detection performance of pelvic MRI for different subtypes of BMinv, we analyzed all pelvic MRI-positive cases in this study. Of the 42 pelvic MRI-positive cases, 19 were also BMB/BMS-positive. As presented in Table 5, only a small proportion (2/16) of pelvic MRI-positive, focal bone marrow lesions were detected using BMB/BMS. Among the pelvic MRI-positive results, 16/42 (38.1%) were of focal lesions. Among pelvic MRI-positive results, only 2/19 (focal/[diffuse + focal] = 10.5%) BMB/BMS-positive results were of focal lesions. This difference in the rate of detection of focal lesions was statistically significant (Fisher exact test: P=0.036).”

Table 5 added to lines 208–209:

Table 5 Pelvic MRI-positive focal and diffuse bone lesions stratified by BMB/BMS results

 Focal bone lesions (cases) Diffuse bone lesions (cases)

Pelvic MRI Positive 16 26

BMB/BMS BMB/BMS (+) BMB/BMS (-) BMB/BMS (+) BMB/BMS (-)

 2 14 17 9

Abbreviations: MRI, magnetic resonance imaging; BMB/BMS, bone marrow biopsy/smear.

Revised text, lines 359–360: “; as expected, a large proportion of pelvic MRI-positive focal bone marrow lesions were not detected using BMB/BMS.”

8. Pelvic MRI, please add years of experience of both readers

Response: Thank you for your comment. I have added this information.

Revised text, lines 97–99: “The physicians who used pelvic MRI for diagnosis were Dr. Dong Xie (13 years of diagnostic experience) and Dr. Zheng Wang (12 years of diagnostic experience).”

9. PET-CT, please add years of experience of both readers

Response: Thank you for your suggestion. I have added this information.

Revised text, lines 96–97: “The physicians who used PET/CT for diagnosis were Dr. Guo-You Xiao (12 years of diagnostic experience) and Dr. Xin Zhao (11 years of diagnostic experience).”

10. PET/CT, I see that you reviewed whole-body PET/CT images, but it is not clear if you considered whole-body PET/CT or only pelvic PET/CT images for the comparison with pelvic MRI. This is crucial and should be clarified. Indeed, reading the Results, in Table 4 I see only pelvic PET/CT locations. Didn’t you consider possible rib, cervico-thoracic spine or other bone locations? This is not consistent with clinical practice and shouldn’t be done in a study aimed to validate pelvic MRI as a diagnostic test in the work-up of lymphoma patients, thus you cannot simply include this point as a limitation of the study. So, even though I understand that common locations are in pelvic bones, if you considered only pelvic bone locations on PET/CT, I strongly suggest to review all PET/CT images to include other potential bone marrow locations in order to assess the actual diagnostic value of pelvic MRI. It should be investigated the clinical and prognostic impact of missing non-pelvic bone locations by using only pelvic MRI, this is essential to support the use of pelvic MRI in clinical practice. Alternatively, it should be underlined that your purpose was to assess the effectiveness of MRI and not of pelvic MRI as a specific diagnostic tool, although limiting the evaluation of MRI and PET/CT to the pelvic bones of course improve the accuracy and agreement of these modalities, reducing the risk of missing other locations (e.g. ribs can be challenging in WB-MRI).

Response: Thank you for your comment. We apologize for not explaining this point clearly in our manuscript. We indeed used whole-body PET/CT for comparison to pelvic MRI. We included these whole-body PET/CT data in our final statistical analysis. The purpose of Table 6 (previously Table 4) is merely to compare the consistency of BMinv detection in the pelvic field between PET/CT and pelvic MRI. To ensure that readers are not confused regarding this point, we added “whole-body” before “PET/CT” in several locations throughout the manuscript.

There were several reasons why we chose pelvic MRI. First, BMB is often performed in the pelvic region, as that is the region in which the bone marrow content is the most abundant. Second, selecting pelvic MRI allows for a better comparison with cytology and pathology results. We discovered that the detection rate of BMinv using pelvic MRI was higher than that using BMB/BMS, and it was in good agreement with whole-body PET/CT. Using whole-body PET/CT, we discovered that only three patients who had BMinv in non-pelvic bone locations did not also have BMinv in the pelvic region.

Whole-body MRI is a good strategy to evaluate BMinv. However, there are several challenges that are hard to overcome. 1: Its resolution is lower than that of pelvic MRI; 2: as you indicated, it is subject to motion artifacts, complicating its use in the diagnosis of BMinv of the thoracic spine and ribs, due to patient breathing; and 3: it is time-consuming, expensive, and not available in all hospitals, which reduces its practicability and patient acceptance.

We are aware that any assay (BMB/PET-CT/pelvic MRI) has limitations and that one should find a balance between sensitivity and specificity in detecting BMinv. The sensitivity of pelvic MRI in the diagnosis of lymphoma with BMinv is better than that of BMB/BMS, and slightly worse than of whole-body PET/CT (pelvic MRI misses the small proportion of patients with independent non-pelvic BMinv). Therefore, considering its cost and accessibility, we think that pelvic MRI is easy, effective, and sufficient for evaluating BMinv.

Revised text, lines 347–351: “However, there are several challenges with whole-body MRI that are hard to overcome. Its resolution is lower than that of pelvic MRI; it is subject to motion artifacts, complicating its use in diagnosis of BMinv of the thoracic spine and ribs, due to patient breathing; and it is expensive, time-consuming, and not available in all hospitals, which limits its clinical application.”

11. BMB/BMS, please add years of experience of both pathologists. Further, you should improve this paragraph including more info about histology and immunohistochemistry analysis.

Response: Thank you for your suggestion. I have added this information.

Revised text, lines 94–96: “The physicians who used BMB/BMS for diagnosis were Dr. Ling-Sha Huang (18 years of diagnostic experience) and Dr. Mei-Qi Li (16 years of diagnostic experience).”

Revised text, lines 132–134: The morphology of lymphoma cells is characterized by a diffuse proliferation of large lymphoid cells, and its immunohistochemistry is characterized as CD20+, CD3−, CD45+, CD79a+, and cyclin D1−.

12. Statistical analysis, I suggest you to assess also the agreement between the three tests, not only to compare the detection rate.

Response: Thank you for this suggestion. I have added ICC analysis to the study.

Revised text, lines 159–163: “Further, we performed intra-class correlation (ICC) in the “irr” R package (version 0.84.1) to estimate intermethod agreement. The limits of agreement were defined as the mean ± 1.96 standard deviations of the difference between a pair of ratings. An ICC value <0.4 was defined as a poor agreement and an ICC value >0.75 was defined as a good agreement.”

Revised text, lines 182–183: “In terms of the consistency of diagnosis using the three methods, Table 3 provides a detailed overview of ICCs.”

Table 3 added to lines 185–188:

Table 3 Intermethod agreement

Comparison ICC [95% CI]

BMB/BMS vs. PET/CT vs. pelvic MRI 0.866 [0.832,0.895]

BMB/BMS vs. PET/CT 0.864 [0.821,0.898]

BMB/BMS vs. pelvic MRI 0.829 [0.775,0.870]

PET/CT vs. pelvic MRI 0.906 [0.875,0.929]

Abbreviations: ICC, intraclass correlation; CI, confidence interval; BMB/BMS, bone marrow biopsy/bone marrow smear; PET/CT, positron emission tomography-computed tomography; MRI, magnetic resonance imaging.

Revised text, lines 363–366: “The consistency between the three methods was good (ICC>0.75), with PET/CT and pelvic MRI exhibiting the highest diagnostic agreement. Therefore, our pelvic MRI procedure may be a viable alternative to other BMinv imaging methods.”

13. Limitations, Line 352, “included only patients with DLBCL”. This is not a limitation, indeed BMB is no longer used in HL, while it will remain essential in marginal lymphoma to evaluate marrow cellularity and hematopoietic reserve. The issue of BMI in especially in DLBCL in which BMI is frequent and often presents as focal rather than diffuse (as in marginal zone), with still unclear role of diffuse FDG uptake on PET.

Response: Thank you for your comment. I have revised the relevant section.

Revised text, lines 415–417: “Further, our study had a retrospective design. Prospective studies with larger sample sizes and with long follow-up periods may help to illustrate the diagnostic value of pelvic MRI in terms of BMinv in patients with DLBCL.”

14. Lines 360-361, I would also underline that you didn’t inject contrast media during MRI, this is another strength as suggested by several authors [Whole body magnetic resonance in indolent lymphomas under watchful waiting: The time is now. Eur Radiol. 2018 Mar;28(3):1187-1193.; Whole-Body Magnetic Resonance Imaging in Oncology: Uses and Indications. Magn Reson Imaging Clin N Am. 2018 Nov;26(4):495-507].

Response: Thank you for this suggestion. I have added this information.

Revised text, lines 430–432: “In any case, MRI is advantageous in its use of non-ionizing radiation, its non-invasive nature, and in its independence of intravenous contrast agents (Galia M, Albano D, Tarella C, Patti C, Sconfienza LM, Mulè A, et al. Whole body magnetic resonance in indolent lymphomas under watchful waiting: the time is now. Eur Radiol. 2018;28: 1187–1193; Petralia G, Padhani AR. Whole-body magnetic resonance imaging in oncology: uses and indications. Magn Reson Imaging Clin N Am. 2018;26: 495–507).”

Reviewer #2: Dear Authors,

This is an article about the diagnostic accuracy of pelvic MRI in BM involvement of DLCBL. Your efforts should be commended and the issue is of relevant interest for the Scientific Community. Also, the manuscript is well written and easily readable.

However, I have a few requests:

1) Please, in the whole manuscript use a different acronym than BMI (i.e., BMinv) because it could be misleading.

Response: Thank you for your comment. I have made the recommended change throughout the manuscript.

2) It would be interesting to add in the figures’ legends, the blood glucose level of each correspondent patient.

Response: Thank you for your comment. I also think it is a good idea, and added the requested information to the relevant figure legends.

3) Page 17 line 295: “PET/CT exposes patients to high doses of radiation”.

Do you refer to PET with diagnostic CT?. Also, you cited a paper of 2014. I would like that you briefly comments about the Digital PET/CT upgrade in terms of radiation exposure (reduction of almost 20% per scan), and other technical opportunities with appropriate references.

Response: Yes, we indeed referred to PET with diagnostic CT. Thank you for the suggestion. I agree that digital PET/CT may improve this situation and I added this information to our paper.

Revised text, lines 331–336: “Moreover, PET/CT exposes patients to high doses of radiation, especially as it may be used repeatedly, such as for staging and post-treatment evaluation. However, digital PET/CT may improve this situation. In a study by Sekine et al (Sekine T, Delso G, Zeimpekis KG, de Galiza Barbosa F, ter Voert EEGW, et al. Reduction of 18F-FDG Dose in Clinical PET/MR imaging by using silicon photomultiplier detectors. Radiology. 2018;286: 249–259), clinically sufficient PET image quality was obtained with simulated reduction of up to 40% of the standard injected dose of FDG when using silicon photomultiplier (SiPM) detectors. Thus, substantially lower radiation doses can be achieved with PET imaging using SiPM detectors compared with those in conventional PET/CT.“

4) Do you consider that the paper’s results are also reproducible for 1.5T MRI scanner? Please, briefly comment on this potential issue.

Response: This is a very important issue concerning the compatibility of the method using different instruments. Unfortunately, we do not have relevant data to answer this question. I have commented on this potential issue as a limitation to our study. We will also consider this aspect in subsequent research.

Revised text, lines 407–414: “One limitation of this study was that we did not evaluate the reproducibility of our pelvic MRI protocol for diagnosing BMinv on 1.5T MRI scanners. In previous studies, it was demonstrated that multiparametric MRI modules, such as iron-corrected T1 and T2*, were repeatable and reproducible across different scanner manufacturers (Philips and Siemens) and field strengths (1.5T and 3.0T) (Bachtiar V, Kelly MD, Wilman HR, Jacobs J, Newbould R, Kelly CJ, et al. Repeatability and reproducibility of multiparametric magnetic resonance imaging of the liver. PLoS One. 2019;144: e0214921; Bane O, Hectors SJ, Wagner M, Arlinghaus LL, Aryal MP, Cao Y, et al. Accuracy, repeatability, and interplatform reproducibility of T1 quantification methods used for DCE-MRI: results from a multicenter phantom study. Magn Reson Med. 2018;79: 2564–2575). They also demonstrated that a large difference in fat content between tissues affects the reproducibility of those MRI modalities. Future studies will be needed to confirm whether 1.5T MRI scanners can be used for accurate pelvic MRI diagnosis of BMinv.”

5) Please, also discuss the potentiality of the obvious synergies of simultaneous PET/MRI in such a scenario.

Response: Thank you for this suggestion. This is a very interesting subject. As Ponisio et al (Ponisio MR, McConathy J, Laforest R, Khanna G. Evaluation of diagnostic performance of whole-body simultaneous PET/MRI in pediatric lymphoma. Pediatr Radiol. 2016;46: 1258–1268) stated, simultaneous PET/MRI “is a promising new modality that combines the metabolic information of PET with superior soft-tissue resolution and functional imaging capabilities of MRI.” PET/MRI is equivalent to PET/CT as a whole-body staging examination in lymphoma patients, with an improved radiation safety profile. I have commented on this issue in the discussion section.

Revised text, lines 336–342: “In addition, simultaneous PET/MRI combines the metabolic information obtained using PET with the superior soft-tissue resolution obtained using MRI. Previous studies have demonstrated that simultaneous PET/MRI is clinically feasible in patients with lymphoma (Atkinson W, Catana C, Abramson JS, Arabasz G, McDermott S, Catalano O, et al. Hybrid FDG-PET/MR compared to FDG-PET/CT in adult lymphoma patients. Abdom Radiol. 2016;417: 1338–1348; Ponisio MR, McConathy J, Laforest R, Khanna G. Evaluation of diagnostic performance of whole-body simultaneous PET/MRI in pediatric lymphoma. Pediatr Radiol. 2016;46: 1258–1268). PET/MRI performance was demonstrated to be comparable to that of PET/CT for lesion detection and measurements of standardized uptake values. Replacement of PET/CT with PET/MRI would markedly decrease the radiation doses to which lymphoma patients are exposed during diagnostic imaging.”

Journal Requirements

Response: Thank you for your advice. I have ensured that my manuscript meets PLOS ONE’s style requirements.

2. We noted that you refer to your original study as a clinical trial through out the manuscript, but according to your description and the WHO definition of clinical trials we would not consider this a clinical trial, since although patients were prospectively assigned to both imaging modalities, there does not appear to have been any analysis of the specific effects of the imaging modality on patient health outcomes, either directly or as a result of differential treatment based on diagnosis. In order to avoid confusion we would suggest that you change the wording in your manuscript and avoid referring to this study as a clinical trial.

Response: Thank you for your advice. I have changed the wording in my manuscript, from “clinical trial” to “study.”

"NO - The funders had no role in study design, data collection and analysis, decision to publish, or preparation of the manuscript."

a.Please clarify the sources of funding (financial or material support) for your study. List the grants or organizations that supported your study, including funding received from your institution.

b.State what role the funders took in the study. If the funders had no role in your study, please state: “The funders had no role in study design, data collection and analysis, decision to publish, or preparation of the manuscript.”

c.If any authors received a salary from any of your funders, please state which authors and which funders.

d.If you did not receive any funding for this study, please state: “The authors received no specific funding for this work.”

Response: We have included the amended statements within our cover letter, as requested. This work was supported by grants from the Guangxi Natural Science Foundation Program (2016GXNSFDA380029, 2018GXNSFBA281026, and 14124003-4) ,Fellowship of China Postdoctoral Science Foundation(2020M673555XB),the Self-financed Scientific Research Project of Guangxi Zhuang Autonomous Region Health and Family Planning Commission (Z2016504) and the Guangxi Science and Technology Foundation (GK2013-13-B-01).The funders had no role in study design, data collection and analysis, decision to publish, or preparation of the manuscript. 

Response: In terms of the data availability, we have uploaded the table with all of the original data that we collected into the Dryad (repository) database (https://datadryad.org/stash/share/vh_5AxZAqKtu_yfz-722Pwf4GibdwIe874qoCHpYi-U). However, there are ethical and legal restrictions in terms of sharing images from the study, as our image data contain potentially identifying or sensitive patient information. The contact information for the Ethics Committee of Guangxi Medical University Cancer Hospital is as follows: tel: +86 771 5330855; email: gxkjk101@163.com.

---

## [Editor Report · Decision Letter 1]

12 May 2021

Diagnostic accuracy of pelvic magnetic resonance imaging for the assessment of bone marrow involvement in diffuse large B-cell lymphoma

PONE-D-20-35498R1

Dear Dr. Li,

We’re pleased to inform you that your manuscript has been judged scientifically suitable for publication and will be formally accepted for publication once it meets all outstanding technical requirements.

Kind regards,

Matteo Bauckneht, MD, PhD

Academic Editor

PLOS ONE
---

## [Editor Report · Acceptance letter]

19 May 2021

PONE-D-20-35498R1 

Diagnostic accuracy of pelvic magnetic resonance imaging for the assessment of bone marrow involvement in diffuse large B-cell lymphoma 

Dear Dr. Li:

I'm pleased to inform you that your manuscript has been deemed suitable for publication in PLOS ONE. Congratulations! Your manuscript is now with our production department. 

Kind regards, 

on behalf of

Dr. Matteo Bauckneht 

Academic Editor

PLOS ONE